# Understanding the Health Behavior Decision-Making Process with Situational Theory of Problem Solving in Online Health Communities: The Effects of Health Beliefs, Message Credibility, and Communication Behaviors on Health Behavioral Intention

**DOI:** 10.3390/ijerph18094488

**Published:** 2021-04-23

**Authors:** Xiaoting Xu, Honglei Li, Shan Shan

**Affiliations:** 1School of Information Management, Nanjing University, Nanjing 210023, China; xxt9337@163.com; 2Department of Computer and Information Sciences, Northumbria University, Newcastle Upon Tyne NE1 8ST, UK; shan.shan@northumbria.ac.uk

**Keywords:** communication behaviors, health beliefs, HPV, intention, message credibility

## Abstract

Online health communities (OHCs) offer users the opportunity to share and seek health information through these platforms, which in turn influence users’ health decisions. Understanding what factors influence people’s health decision-making process is essential for not only the design of the OHC, but also for commercial health business who are promoting their products to patients. Previous studies explored the health decision-making process from many factors, but lacked a comprehensive model with a theoretical model. The aim of this paper is to propose a research model from the situational theory of problem solving in relation to forecasting health behaviors in OHCs. An online questionnaire was developed to collect data from 321 members of online health communities (HPV Tieba and HPV vaccina Tieba) who have not received an HPV vaccination. The partial least squares structural equation modeling (PLS-SEM) method was employed for the data analysis. Findings showed that information selection and acquisition is able to forecast HPV vaccination intentions, perceived seriousness and perceived susceptibility can directly impact HPV vaccination intention and have an indirect impact by information selection and acquisition, and perceived message credibility indirectly affected HPV vaccination intention via information selection. The current paper supports health motivations analysis in OHCs, with potential to assist users’ health-related decision-making.

## 1. Introduction

Online health communities (OHCs) rise up as a new type of tool to regulate people’s health behavior and decision-making process. People are now dealing with their health and chronic conditions differently, due to social media and related Web 2.0 technologies. These individuals search the Internet to seek information and advice on their conditions, as well as alternative treatments and other patients who are dealing with the same problems [1]. In line with this, OHCs are a critical platform with which patients are able to connect with others suffering from similar conditions, share information and experiences, and connect with healthcare professionals [2]. OHCs offer users the opportunity to find reliable information related to particular ailments and unique treatments and share their experiences with fellow users [3,4]. Users are able to share and create health information anonymously without worrying about their privacy due to the security design of the website [5]. Consequently, people’s health behavior and decision-making regarding certain diseases are greatly shaped and influenced by the OHC contents. For example, a woman diagnosed with breast cancer will seek knowledge about the treatment options, choice of drugs for treatment, and surgery options from OHC, which will shape her health belief on each choice of treatment, leading to a final decision on which drugs to use and a surgery decision.

Understanding what factors influence people’s health decision-making process is essential for not only the design of the OHC, but also for commercial health businesses who are promoting their products to patients [6,7]. This particular area of interest has become the focus of extensive research in the contemporary eras [8,9]. Earlier research has shown that the health beliefs of people are important in relation to persuading people to alter any unsuitable behaviors and move onto healthier habits [10,11,12]. Certain researchers showed that perceptions a person had about health (including perceived susceptibility, perceived seriousness, perceived benefits, and perceived self-efficacy) are able to forecast their beliefs and future decisions related to health behaviors [10,13]. Even though health behavior research has been successful, there are at least two research gaps. Firstly, the perceptions and communication activities of users were not taken into account. This is seen where human behavior is encouraged by online communities, and media perception was overlooked, particularly for message source perception. Current research has shown that message credibility increases the distribution of information, information seeking, information selection, and appraisal [14,15]. As the foundation for health communication of OHCs and personal health behavior, message credibility perception is a topic that should receive extensive research attention. The second gap is not taking the public’s varying communication behaviors into account. Using the situational theory of problem solving (STOPS), communication behavior involves the collection, selection, and dissemination of information [8,9]. On the other hand, research examining the link between communication behavior and health behavior is limited.

The current study intends to fill these two gaps through analyzing the way individuals’ health and message credibility perceptions and communication behaviors impact their health behaviors in online health communities. Specifically, OHC users with an interest in the human papillomavirus (HPV) vaccine are used as the study sample. This particular vaccination carries a significantly high efficacy in HPV infection prevention, therefore receiving attention from across the globe. On the other hand, there is limited public acceptance of this vaccine, even though it has had positive results [16]. The CDC reported that only 51% of all teens in the USA that had been administered the suggested doses of HPV vaccine received all recommended doses of the HPV vaccine [17]. While the vaccine was approved in 2016 in China, poor levels of public awareness (16%) and acceptability (67%) have restricted the vaccine’s promotion [18]. Therefore, the current paper intends to establish a model in OHCs with which to forecast the health behaviors of users with regards to perceptions and communicative behavior, and then investigate the importance of communication behaviors in perceptions and health behaviors.

There are four key contributions made by this study towards comprehending health behaviors in OHCs. These include creating a new model with which to forecast health behavioral intentions, presenting the way communication behaviors can positively impact health behaviors, and confirming that perceived risk has a direct and indirect impact on health behaviors and communication behaviors, respectively. Notably, perceived message credibility will be shown to exclusively impact health behaviors indirectly by a mediating variable of communication behaviors. The final contribution of this study is to assist in the regulation of information quality, and offer users a secure and active communication environment, acting as a health campaign that takes into account people’s health perceptions as well as the media perception and media credibility of messages. The paper is organized as follows. Section 2 presents the literature review and research hypotheses, Section 3 explains the research methodology for the paper, Section 4 presents the data analysis results of the study, and Section 5 discusses the results and implications of the study, followed by the conclusion of the study in Section 6.

## 2. Conceptual Background and Research Hypotheses

### 2.1. Communication Behaviors and Situation Theory of Problem Solving

Human beings are highly social creatures who act in socially structured environments where communication plays a key role. For people to share information with each another in public space, communication behavior can be explained by the situational theory of publics (STP), under which individual communication behavior and decision-making can be seen as a life problem solving process through which individuals actively seek and process information [19,20]. Kim and Grunig [21] have further classified communication behaviors into the three main categories, information acquisition, information selection, and information transmission, and extended STP into the situational theory of problem solving (STOPS). Shen, et al. [22] showed that STOPS can specify the independent variables of the theory of publics for communication action in problem-solving. In turn, these independent variables are defined as recognition of problem, recognition of constraint, recognition of involvement, and elements involved in the criteria for referral. With STOPS, the communication behavior is explained through the desire to solve problems. When an individual has a problem to solve, he or she will be motivated to seek information relevant to the problem. For this reason, STOPS is applied to explain communication behavior to support that an individual seeks an optimal solution for the problem based on the independent variables involved [23]. As an individual experiences a growing number of issues or problems in their life, their communication behaviors will become more active. The three variables above also relate to each other. The greater the problem’s solution, the greater amount of information related to the problem will be collected and the greater selectivity in handling the information, along with increased transmission to others [21]. Based on the STOPS theory, we proposed that the communication behavior will influence the behavioral intention for the actual behavior of the problem, i.e., if an individual seeks information related to breast cancer surgery, her behavioral intention of having breast cancer surgery will be influenced by the information selection, information transmission, and information acquisition of breast cancer surgery. Additionally, the communication behavior is influenced by the health belief and message credibility. The conceptual model is presented in Figure 1.

Information selection is vital when it comes to information transmission for effective communication, and through technological advances, consumers have more new communication tools at their disposal. Information is trusted by consumers when it is of the intended quality. Stephens and Mandhana [24] showed that information transmission is able to be improved or worsened based on the media selected, and, ideally, the media type used should correctly transmit the messages and allow for both effective communication and reception of information. Therefore, those who constantly disseminate, search for, and collect information will continue to search for new information that others share [25].

It should also be noted that media selection for information transmission can have a positive effect on the recipient’s information acquisition. Information transmission is facilitated by selected media, and quality transmission of information has a correlation with the quality of information acquisition. Kulahci, et al. [26] put forward the notion that the recipient’s information acquisition is closely related to the transmission. This means that it is necessary for the information transmitter to consider the nature and quality of the media chosen for the transmission of messages. In turn, choice of transmission system has an effect on the message the recipient is presented with [27]. A cervical cancer prevention campaign was conducted on Twitter, through which Yoo, et al. [28] confirmed their suggested link via empirical research. Information recipients choose information related to their needs and its relevance [29]. As people are presented with an overwhelming amount of data on a constant basis, selectivity is necessary for greatest effectiveness, and thus communication behavior is affected. Information acquisition has a substantial effect on decision making, which in turn decides the recipient’s desired information to be selected [30]. Information selection is established through the media used for information acquisition, and so the following hypotheses are put forward:

**H1.** *Information selection, information transmission, and information acquisition will have positive relationships*.

**H1-1.** *Information selection will have a positive relationship with information transmission*.

**H1-2.** 
*Information transmission will have a positive relationship with information acquisition.*


**H1-3.** *Information selection will have a positive relationship with information acquisition*.

There have been a number of studies using STOPS in the context of behavioral intention research, in order to examine how communication behavior and behavioral intention are linked. Villafranca, et al. [31] state that communication problems can occur in hospital environments because of disruptive activities. It is necessary for suitable communication behaviors to be present at health facilities in order to produce ideal health intentions. The work of Rosenstein [32] showed that disruptive behaviors can put pressure on staff relationships and could restrict communications. Medical teams that lack coordination and have poor collaboration with others can bring about negative results to their patients requiring intense care [32]. Strong communication skills and sufficient levels of respect in health workplaces reduce disruptive behaviors, therefore facilitating greater patient safety [33]. Effective communications can bring about positive reception and selection of information for effective response, in line with the receiver’s needs. The work of Sultan, et al. [34] found that effective communications can bring about positive perceptions of the messages, as intended, therefore encouraging the intended activities. Information is more likely to be selected by patients when they feel it would benefit their conditions. Additionally, Kim, Shen and Morgan [25] showed that when an individual’s communication behavior was more positive regarding organ donation, then their donation behavioral intention would also be greater. Rimal [35] put forward the notion that when individuals have enough information on a desired topic, they will be in a stronger position to instigate the necessary behavioral changes. Lee and Rodriguez [36] showed that individuals who actively looked for bioterrorism information had greater preference for taking actions throughout bioterrorism attacks (before, during, and after). In line with this, research into tuberculosis infections in Mexican Americans showed that individuals with greater focus on media cues also had stronger intentions to take part in TB health behaviors [37]. Shi, et al. [38] investigated the properties related to middle-aged Japanese men’s health habits changes and presented the finding that consciously looking for health information had a high correlation with positive changes in health behaviors. Additionally, through a cervical cancer prevention campaign, it was found that information acquisition had a positive correlation with HPV vaccination Yoo, Kim and Lee [28].

Notably, Di Virgilio and Antonelli [39] stated that media selection for content sharing is a critical component of patient information selection, and is directly linked with behavioral action. The medium employed in these transmissions must be user-friendly and accessible for the target audience decided by the hospitals. In turn, the intended behavioral response is facilitated when the appropriate (and well respected) media are chosen [40]. Therefore, patients who cannot communicate easily need suitably simple media in order to achieve best action by the patient as well as the medical team providing health care. Burgener [41] stated that poor communication exposes patients to greater levels of risk. This is evident in cancer patients, who require specific care in order to manage their treatment period. Ineffective communication induces stress and harm to patients who are struggling physically and mentally in other ways as well. In turn, appropriate communication between nurse and doctor will facilitate more suitable actions being taken, therefore helping their patients recover quicker. Conversely, poor communication from medical staff brings about a lack of coherence and negative evaluations. Thus, the hypothesis below is proposed:

**H2.** *Communication behaviors will have a positive relationship with behavioral intention*.

**H2-1.** *Information selection will have a positive relationship with behavioral intention*.

**H2-2.** *Information transmission will have a positive relationship with behavioral intention*.

**H2-3.** 
*Information acquisition will have a positive relationship with behavioral intention.*


### 2.2. Health Beliefs

In the 1950s, the health belief model (HBM) was developed, which uses social psychology aspects when assessing behavioral decisions related to individual health and disease prevention [42]. This is considered to be a critical component of forecasting healthy behavioral intentions [43,44]. In the beginning, the model involved four concepts, namely perceived susceptibility, perceived seriousness, perceived benefit, and perceived barriers [10]. This was then extended to include health motivation, self-efficacy, and others [13,45,46]. Once the health belief model was put forward, it received extensive application in health behavior sectors in order to understand health education and health intervention [10,47,48]. Green, et al. [49] showed that, using the Health Belief Model and the theory of subjected expected utility, health threats to patients must be explained clearly, for the individual to feel hopeful about their subsequent recovery. The potential dangers and seriousness of the patients’ health conditions must be made conditional considering the potentially superior care they could receive. As a result of Covid-19, health practitioners must encourage behaviors which would restrict the growth of the virus, through using the health belief model [50].

In addition, a large number of researchers have shown that HBM is effective in describing breast self-examination, vaccinations, exercise, physical activity, smoking, seat belt use, gestational diabetes mellitus, and other health-related behaviors [11,43]. Furthermore, earlier research has stated that health beliefs have positive correlations with communication behaviors. In the work of Rimal and Real [51], the risk and efficacy perception of skin cancer people had would dictate their information seeking activities. Additionally, Kaphingst, et al. [52] showed that individuals with higher perceived risk had a greater likelihood to seek out cancer information. On the other hand, limited knowledge regarding breast cancer means that more effective communication is required. Deeper knowledge would allow for more efficient actions to be taken. In turn, specific patients who are at risk of particular diseases have a greater likelihood of denying their risk, including cancer, TB and AIDS [53]. The work of Green, Murphy and Gryboski [49] puts forward the notion that belief systems affect the perceptions people have regarding medical care, as well as their outlook regarding specific diseases.


***Health beliefs and behavioral intention***


There are two key variables of perceived risk under the health belief model, which are perceived susceptibility and perceived seriousness [10,54]. Research has shown that perceived seriousness towards melanoma has a substantial impact on sun-tanning behavior [11]. On the other hand, perceived susceptibility shows a positive correlation with attitudes and perceptions related to injury prevention programs [12]. Patients at risk of a disease are in a stronger position to make decisions if they are privy to important data to select from. Furthermore, Chen, et al. [55] noted that individuals with access to information related to disease symptoms, availability of drugs, and medical care have a greater chance to act in a preventative manner, compared to individuals who lack this information [28]. Access to information allows a sense of self-efficacy to encourage people to take the required steps [49].

Information acquisition is a vital aspect of decision-making for people who are at risk of disease, to receive adequate medical care. When establishing perception, access to information and knowledge is significant, and the same applies when it comes to susceptibility of patients. When knowledge dissemination is more effective, then people will have a more educated outlook towards the pandemic. Social media allows rapid communication, which allows for in-depth information acquisition, leading to preventive behaviors [56]. Therefore, effective information acquisition can bring about the proposed behavioral change. In cases where households saw that there are serious consequences from zoonotic infections, there was greater intention to manage pests [57]. In addition, a number of studies employ the theory of fear appeals, in order to clarify behavioral decisions, showing that a strong link exists connecting perceived susceptibility and perceived seriousness with fear [58,59]. It cases where people perceive a risk, or a fearful situation, they are more likely to make behavioral changes [60]. In turn, once a person perceives they are at greater risk of the flu, and that the flu would have severe repercussions, their willingness to be vaccinated increases [61]. Additionally, it was shown that the perceived susceptibility and perceived seriousness of individuals had a clear impact on their outlook and intention regarding HPV vaccination [28].

Yoo, Kim and Lee [28] also showed that information, which is mediated via social media, allows for intentional behavior to change, through communication and information acquisition. In addition, Kim, et al. [62] put forward the notion that individuals who must deal with contradictory data about their current health beliefs attempt to find new information about their health. In turn, transmission of information to these individuals influences susceptibility and decision-making related to protecting their health. These research findings lead towards the belief that perceived susceptibility and perceived seriousness can affect behavioral intention. Thus, this research suggests hypothesis H3:

**H3.** 
*Health beliefs will have positive relationships with behavioral intention.*


**H3-1.** *Perceived susceptibility will have a positive relationship with behavioral intention*.

**H3-2.** *Perceived seriousness will have a positive relationship with behavioral intention*.


***Health beliefs and communication behaviors***


Research has shown that individuals who have higher perceived risk will make efforts towards information acquisition, selection, and transmission action. Once these individuals feel they are susceptible to a particular ailment, they acquire positive information seeking behavior [35]. In the study by Lee and Rodriguez [36], it was denoted that people who perceive bioterrorism to be a vital social issue will be more likely to actively acquire and process information about this topic. Additionally, Griffin, et al. [63] put forward a new model of risk information seeking and processing, showing that an individual’s perceived risk encourages information seeking and processing. Along the same lines, Shakeri, et al. [64] produced analogous results, where their scenario-based online survey of college students showed that perceived risk directly impacted information seeking motivation. Research into food safety highlighted that risk perception was a powerful predictor of information need and information-seeking intention [65]. Lastly, Lee, et al. [66] put forward the notion that expected risk had a clear impact on users’ intention to share through social media. These research outcomes suggest that health belief has a correlation with three variables of communication behaviors, while still not being unified. Therefore, hypothesis H4 is suggested:

**H4.** *Health beliefs will have positive relationships with communication behaviors*.

**H4-1.** *Perceived susceptibility will have a positive relationship with information selection*.

**H4-2.** *Perceived susceptibility will have a positive relationship with information transmission*.

**H4-3.** *Perceived susceptibility will have a positive relationship with information acquisition*.

**H4-4.** *Perceived seriousness will have a positive relationship with information selection*.

**H4-5.** *Perceived seriousness will have a positive relationship with information transmission*.

**H4-6.** *Perceived seriousness will have a positive relationship with information acquisition*.

### 2.3. Message Credibility

A significant amount of research has highlighted the connection between message credibility and variables of communication behavior. In the work of Yan, Zhou, Wang and Li [15], WeChat users in China were analyzed, and the findings showed that information credibility has a direct and positive impact on information dissemination in the context of WeChat. Furthermore, Dedeoglu (2019) presented the finding that the credibility perceptions of tourists towards social media had a positive correlation with the value they gave non-participant shared content [67]. Another study into obesity public service announcements (PSAs) put forward the notion that in situations where spokespeople in advertisements are real people instead of actors, the information seeking tendencies of overweight viewers will be stimulated, as they feel that what they say is genuine [14]. Certain studies into the online health information seeking tendencies of older adults showed that credibility is the key factor in deciding to search for online health information [68]. For most of the public, information selection is more frequent from news sources that have a strong reputation [69]. In addition, credibility has an extra effect on the various phases of information selection and evaluation, and it is particularly impactful during page switching [70]. These findings can also be applied to economics, where credibility supports the development of the customer attitude regarding web advertising, additionally affecting web advertising usage for information acquisition [71]. Once investors have collected information, credibility becomes important [72]. These findings produce the following hypothesis:

**H5.** *Message credibility will have positive relationships with communication behaviors*.

**H5-1.** *Message credibility will have positive relationships with information selection*.

**H5-2.** *Message credibility will have positive relationships with information transmission*.

Figure 1 summarizes the conceptual model analyzing the way different factors, such as health beliefs, message credibility, and communication behaviors, affect health behavioral intentions related to HPV vaccination.

## 3. Research Methodology

This paper mainly used the online survey to gather data to validate our theoretical model. The online survey was used because it is convenient to identify people who are interested in HPV vaccination.

The partial least squares structural equation modeling (PLS-SEM) was used to develop the conceptual model. This is considered to be an appropriate tool in exploratory research and theoretical development and particularly in evaluating numerous variables in a complex model [73,74]. The current paper uses the SmartPLS 3.3.2 software (SmartPLS GmbH, Bönningstedt, Germany) [75]. The Path Weighting Scheme estimation method is used, while significance calculations are achieved through bootstrapping when using Smart PLS in order to appraise the model. Hair, Ringle and Sarstedt [74] recommended that t-statistics are computed using 5000 bootstrap samples, and this suggestion was implemented in the current context. Model estimation consisted of a two-step methodology, with the measurement model first and then the structural model second.

### 3.1. Sample and Procedure

This paper is primarily concerned with people who have a level of concern regarding HPV-related information, and who have not been vaccinated against HPV. It is considered that the epidemiology of cervical cancer is critical to its prevention [76]. In turn, HPV stands as the etiological element and biologic carcinogen for HPV-associated lesions and cancers [77]. As a result, finding HPV to be the etiological factor for HPV-associated malignancies allows for greater control of these cancers with vaccinations, and other therapeutic strategies [78]. The HPV condition was chosen as it has been defined as a sexually transmitted virus, and this means that people are more likely to communicate online and anonymously about the condition.

Empirical data were gathered through inviting people to take part in a survey, from March to June 2020, through the “HPV Tieba” and “HPV vaccina Tieba” subordinates to Baidu Tieba, which is the biggest online community in China (http://c.tieba.baidu.com/f?ie=utf-8&kw=hpv&fr=search&pn=0& accessed on 1 March 2020 and https://tiebac.baidu.com/f?kw=hpv疫苗&ie=utf-8 accessed on 1 March 2020).

The data collection questionnaire was published in a professional data collection website, where the repeated identity is avoided through filtering same email addresses and IP addresses, etc. Participants were motivated to take part in the survey by being presented with an overview of the results automatically generated by the survey hosting website after participants submit their completed surveys. There were also random rewards awarded to people who completed valid surveys, as an additional incentive.

### 3.2. Measures

The variables used in this paper were taken from earlier studies, and subjects gave ratings on a seven-point Likert scale. These included the following: (1) intention regarding HPV vaccination, measured using two items adapted from the work of Fishbein and Ajzen [79]; (2) perceived susceptibility and perceived seriousness, evaluated with four items seen in [80,81]; (3) message credibility, as taken from Appelman and Sundar [82] using the semantics of accurate, authentic, and believable; and (4) Information selection, information transmission, and information acquisition evaluated with two, three, and two items, respectively, adapted from [21]. Table 1 displays the items used to evaluate the latent constructs. To maintain the highest level of validity and reliability in the survey, as well as avoid semantic issues, a pre-test was conducted with 23 active HPV Tieba users participating. Suitability of items used was confirmed by removing items with a factor loading under 0.70, through principal component analysis [83]. In turn, item 5 (I worry a lot about being infected with HPV) of perceived susceptibility was removed, as well as item 4 (being infected with HPV will lead to a hopeless diseas) of perceived seriousness.

## 4. Results

### 4.1. Respondents

There were 88 incomplete invalid responses, or responses that included the same response option in most question items, leaving 321 valid responses. Table 2 presents the sample profile. The survey participant sample was 20.25% male and 79.75% female, with most respondents being between 30 and 39 years old (45.79%), with tertiary education (57.63%). As the sample was demographically homogeneous, these variables were excluded from additional analysis as control variables.

### 4.2. Non-Response Bias

Armstrong and Overton suggested that late respondents are more likely to resemble non-respondents than early respondents [84]. To ensure the consistency of the samples, we conducted a non-response bias test. This study tests non-response bias by comparing the gender and age of early respondents to the later ones. A total of 200 respondents who completed the survey during the early stage were considered group 1, and 121 respondents who completed the survey during the later stage were considered group 2. The Chi-Square test for two groups shows that they did not differ significantly (*p* > 0.05) in gender or age. We therefore excluded the possibility of non-response bias.

### 4.3. Common Method Bias

Common method bias is an artificial covariation between the predictive variables and the criteria variables caused by the same data sources, the same measurement environment, context, and the characteristics of the project itself [85]. The common method bias test is to test whether such covariation exists or not; especially in the cross-sectional survey, there is more common method bias in self-reported data collection. In this research, we assessed the data set using Harman’s single-factor method. After exploratory factor analysis, the explanatory percentage of variance of the first common factor is less than 40%, indicating there is no common method bias. We extracted five principal components, the total variance was 62.530%, and the first principal component explains 25.976% of the total variance, less than 40%. Therefore, there is no single principal component that explains most of the variance, indicating that common method bias may not be a serious problem in the data set.

### 4.4. Evaluation of Measurement Model

The current paper employs exploratory factor analysis to ensure the measurement model’s accuracy and authenticity. Earlier researchers have suggested certain reference criteria, namely Cronbach’s alpha (ɑ > 0.70), composite reliability (CR > 0.70), average variance extracted (AVE > 0.50), and discriminant validity [73,74,86]. Cronbach’s alpha can denote the internal consistency of the measurement model, while CR evaluates the questionnaire items’ reliability, and AVE describes convergent validity. Furthermore, discriminant validity was employed to confirm if latent model variable correlation was statistically disparate, and which of the criteria has a square root of AVE higher than the absolute value of the Pearson correlation coefficient of inter-construct. Table 2 presents the item loadings, and it is seen that all values were above the recommended level of 0.7, and significant [74]

To ensure there is sufficient discriminant validity in the latent constructs, square roots of AVE and latent variables were appraised, with results shown in Table 3 [86]. It is seen that all AVE square roots are above the correlations across latent variables pairs, and so discriminant validity is confirmed. In addition, the correlation coefficient (β < 0.6) and all VIF (VIF < 10) between independent variables satisfies the criteria (Bagozzi, 1981), indicating that there is no multicollinearity in our research.

### 4.5. Hypotheses Testing

In Figure 2, the model estimation findings regarding direct impacts amongst constructs are shown. It is seen that aggregate PLS path coefficients have statistical significance. In turn, the relationships across communicative behavior variables (such as information selection, information transmission, and information acquisition) were tested under H1. It was found that a significant positive relationship existed between information selection and information transmission (β = 0.244, *p* = 0.001 < 0.01), and information selection and information acquisition (β = 0.149, *p* = 0.018 < 0.05). However, there was no significant relationship between information transmission and information acquisition (β = 0.073, *p* = 0.166 > 0.05), As a result, H1-1 and H1-3 could be confirmed, while H1-2 was not.

Next, H2 was used to evaluate how communication behaviors and intention towards HPV vaccination were linked. It was found that there was a significant positive relationship between information selection and intention towards HPV vaccination (β = 0.434, *p* = 0.000 < 0.001), and information acquisition and intention towards HPV vaccination (β = 0.164, *p* = 0.000 < 0.001). However, there was no such relationship found for information transmission and intention (β = −0.010, *p* = 0.822 > 0.05). As a result, H2-1 and H2-3 were accepted, while H2-2 was not.

Following this, the relationships between health beliefs (such as perceived susceptibility and perceived seriousness) and intention towards HPV vaccination were examined in line with H3. It was seen that perceived seriousness had a positive correlation with intention towards HPV vaccination (β = 0.142, *p* = 0.002 < 0.01). Similarly, there is a statistical significance for perceived susceptibility and intention (β = 0.217, *p* = 0.000 < 0.001). Thus, H3-1 and H3-2 were accepted.

Under H4, the relationships across health beliefs and communication behaviors were tested, and it was seen that while perceived susceptibility had a significant positive relationship with information selection (β = 0.175, *p* = 0.001 < 0.01) and information acquisition (β = 0.141, *p* = 0.009 < 0.01), there was no statistical significance for information transmission (β = −0.082, *p* = 0.191 > 0.05). In addition, perceived seriousness had a significant positive relationship with information selection (β = 0.270, *p* = 0.000 < 0.001) and information acquisition (β = 0.231, *p* = 0.000 < 0.001), but information transmission lacked statistical significance (β = −0.118, *p* = 0.088 > 0.05). In turn, H4-2 and H4-5 were not accepted, while H4-1, H4-3, H4-4, and H4-6 were supported.

Next, H5 examined how message credibility and communication behaviors were related, finding that message credibility had a positive link with information selection (β = 0.305, *p* = 0.000 < 0.001), but information transmission lacked statistical significance (β = 0.107, *p* = 0.105 > 0.05). Therefore, H5-1 was accepted, while H5-2 was not.

Lastly, factors that forecast information selection, information transmission, information acquisition, and behavioral intention could account for 30.7%, 7.1%, 17.2%, and 49.8% of the variance, respectively.

## 5. Discussion and Conclusions

### 5.1. Major Findings

The current paper looked at the different effect user perceptions, such as perceived susceptibility, perceived seriousness, and perceived message credibility, and communication behavior have on their intention towards receiving HPV vaccination. Earlier research efforts have primarily concentrated on the way that the health beliefs model impacts the decision-making and health behaviors of different individuals [10,11]. However, these scholars gave much less attention to what the message perceptions and specific communication behaviors of these people were prior to making any HPV vaccination decisions. In addition, they overlooked the way their intentions towards HPV vaccination were affected by these factors. In turn, this research study involved users who participate in the “HPV Tieba” and “HPV vaccina Tieba” groups as its empirical object, with PLS used to verify the model regarding the impact of individuals’ perceptions and communication behaviors on their intention towards HPV vaccination, as well as their internal relationship.

It was seen that communication behaviors bring about positive inner relationships, and that information selection and acquisition can forecast the intentions people have related to vaccination. On the other hand, previously published studies concentrate solely on information seeking [19,20,21], with little attention given to the wider communication behaviors of people, including information selection, information transmission, and information acquisition. Considering this, the current paper underlines the significant value the public’s activeness of communication behaviors has in these matters, as it stimulates their health behaviors. This has been put forward in certain earlier studies [51]. This finding is notable, since it suggests that information selection of online health communities can have an effect on health behaviors, as well as stimulate a greater level of information acquisition efforts through other channels and thus impact health behaviors. This can be considered a reason why people have a higher level of trust towards Tieba, while other individuals might require input from numerous channel sources in order to form their final decisions [87].

Furthermore, it can be suggested that perceived seriousness and perceived susceptibility can directly impact intention towards HPV vaccination, which is in line with earlier research [11,57]. Additionally, perceived seriousness and perceived susceptibility have an indirect impact on intention towards HPV vaccination by communication behaviors. We found information selection and acquisition play a mediating role between perceived susceptibility and intention regarding HPV vaccination, and information selection and acquisition mediated the effect of perceived seriousness and intention.

Lastly, the study has shown the way media perception and message credibility affect health behavioral intention. Message credibility was found to have indirect effects on HPV vaccination intentions with communication behaviors, and this is in line with the findings of earlier research that confirmed message credibility has a direct and positive effect on information behaviors [14,15,67]. There is greater willingness to look for information if its credibility in the online health community is greater. Therefore, perceived message credibility increases information selection, which means that information selection will boost health behavioral intentions as well.

### 5.2. Implication for Research

The current paper carries several theoretical implications. Firstly, it is suggested that a new model should be developed to effectively forecast the health behavioral intentions of people. Earlier research on this topic had been limited to suggesting that the perceived risks of people would have an impact on their behavioral intentions [10,11,12]. Additional effects of other perception and communication behaviors were not considered. Therefore, the current paper builds on the health belief model, and involves perceived message credibility and communication behavior into its analysis, which will allow for a deeper understanding of the matter. Specifically, the way health perception (including perceived susceptibility and perceived seriousness), media perception (perceived message credibility), and communication behavior will be able to forecast the future health behavioral intentions of people and their internal relationship will be more clearly comprehended.

A second key implication is that communication behaviors have been shown to increase health behaviors. Earlier studies concentrated primarily on information seeking behavior [20], and certain scholars found that there were a number of information behaviors that affect health behaviors in people [21]. Therefore, the current study decided to examine the communication behaviors from three distinct levels, in line with STOPS. These were associated with the selection, transmission, and acquisition of information. In turn, it was shown that several communication behaviors have internal connections, and that active information selection behaviors in internal OHCs can accurately foresee health behaviors. Furthermore, it is possible that they can motivate additional information behaviors stemming from external network platforms, which would then also affect health behaviors.

A third implication of this study is that perceived risk has a direct impact on health behaviors, while also having an indirect effect as a mediating variable of communication behaviors. However, perceived message credibility was shown to exclusively impact health behaviors in an indirect manner, as a mediating variable of communication behaviors. As a result, it is concluded that the varying perceptions people have will affect health behaviors towards different directions. It should also be noted that scholars have concentrated on the relationship between message credibility and information behaviors to a great extent [14,15]. Therefore, the current results supported the notion that perceived message credibility, communication behaviors, and health behaviors are connected.

### 5.3. Practical Implication

This paper’s results offer several important findings for health communicators. Firstly, health campaigns will realize people’s health perception alongside their media perception, and particularly their feelings regarding the message credibility of media. In turn, health communicators must implement strict control over the information quality, while offering users a secure and active communication environment.

Furthermore, the health decision-making of users is based on the information acquisition from external sources as well as the information selection of the media employed. Therefore, creating links to relevant channels, such as hyperlinks and chat platforms, is important for the user’s decision-making.

### 5.4. Limitations and Future Work

It should be noted that there are certain limitations to this paper. The first limitation is that only two specific perceived risk variables were used in the health belief model, perceived susceptibility, and perceived seriousness. This means that additional investigation is needed to examine if other perceived variables, such as perceived benefit and perceived barriers, can be added to the model to cause different effects on behavioral intention to change. The second limitation is that only perceived message credibility related to media perception was examined, while other perceived utility and subjective norm impacts were not addressed. Future study may consider including other variables such as perceived utility and subject norms into the model. Thirdly, situational experiments were employed to evaluate the intentions of participants, meaning that actual behaviors could not be explained, as only cross-sectional decisions of that moment were made clear. Further studies are recommended, involving a combination of time series data to effectively analyze long-term behavior. The fourth limitation is that although our common bias method is provided, we had limited time to provide the variance inflation factor due to the software not being provided, and we had limited time to calculate it.

## 6. Conclusions

With the improvement of people’s perceptions, users share and seek health information through online health communities (OHCs). This research proposed a model based on the situational theory of problem solving to examine influential factors associated with intention regarding HPV vaccination. These results show that information selection and acquisition are able to forecast HPV vaccination intentions, perceived seriousness and perceived susceptibility can directly impact HPV vaccination intention and have an indirect impact by information selection and acquisition, and perceived message credibility indirectly affected HPV vaccination intention via information selection.

## Figures and Tables

**Figure 1 ijerph-18-04488-f001:**
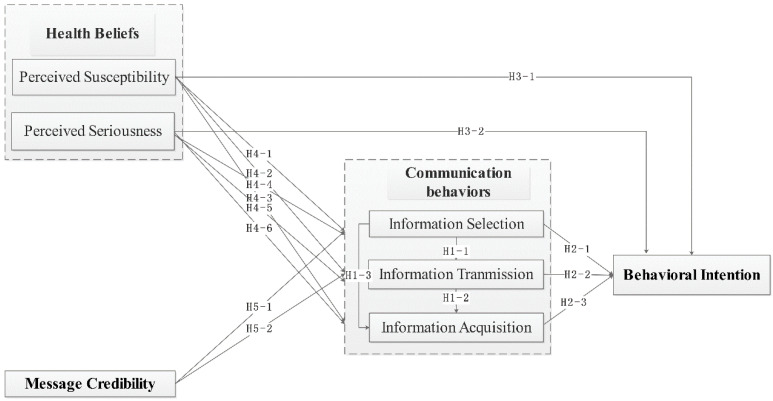
Conceptual model.

**Figure 2 ijerph-18-04488-f002:**
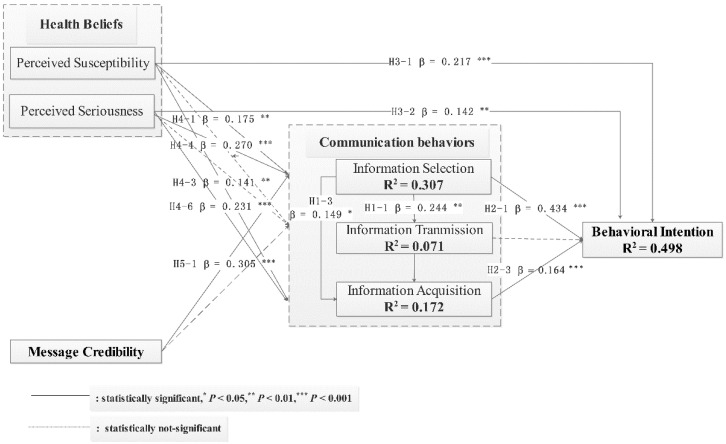
Research Model Test Results.

**Table 1 ijerph-18-04488-t001:** Scale items and convergent validity.

Constructs/Items (7-Point Scale)	Loading
Perceived Susceptibility (CR = 0.855; AVE = 0.597, Cronbach’s = 0.775)	
PSY1 1My chances of being infected with HPV are great.	0.814
PSY2 My physical health makes it more likely that I will be infected with HPV.	0.737
PSY3 I thought I could have HPV.	0.749
PSY4 I feel that my chances of being infected with HPV in the future are great.	0.787
Perceived Seriousness (CR = 0.835; AVE = 0.559, Cronbach’s = 0.736)	
PS1 I believe that the HPV is extremely harmful.	0.701
PS2 The thought of being infected with HPV scares me.	0.777
PS3 HPV is a serious disease that can kill me.	0.798
PS4 If I had been infected with HPV, my career/life/study would be affected.	0.709
Message credibility (CR = 0.848; AVE = 0.650, Cronbach’s = 0.732)	
MC1 The information of Baidu Tieba is accurate.	0.798
MC2 The information of Baidu Tieba is authentic.	0.853
MC3 The information of Baidu Tieba is believable.	0.765
Information selection (CR = 0.930; AVE = 0.869, Cronbach’s = 0.849)	
IS1 How often did you selectively seek HPV-related information from Baidu Tieba?	0.934
IS2 How often did you selectively read HPV-related information from Baidu Tieba?	0.930
Information transmission (CR = 0.888; AVE = 0.726, Cronbach’s = 0.811)	
IT1 How often did you share HPV-related information from Baidu Tieba?	0.857
IT2 How often did you forward HPV-related information from Baidu Tieba?	0.858
IT3 How often did you retweet HPV-related information from Baidu Tieba?	0.840
Information acquisition (CR = 0.877; AVE = 0.780, Cronbach’s = 0.719)	
IA1 How often have you acquired HPV-related information from other websites?	0.871
IA2 How often have you searched for HPV-related information on other websites?	0.895
Behavioral intention (CR = 0.895; AVE = 0.810, Cronbach’s = 0.766)	
BI1 I want to receive the HPV vaccine.	0.915
BI2 I am willing to receive the HPV vaccine.	0.884

Notes: CR = composite reliability, AVE = average variance extracted.

**Table 2 ijerph-18-04488-t002:** Profile of the participants.

Measure	Items	Frequency	%
Gender	Male	65	20.25
	Female	256	79.75
Age	18- 19	38	11.84
	20–29	92	28.66
	30–39	147	45.79
	40 and over	44	13.71
Education	Bachelor	185	57.63
	Master	78	24.30
	PhD	12	3.74
	Others	46	14.33
Profession	Working full time	179	55.76
	Freelancer	35	10.90
	Unemployment	23	7.17
	Full time student	84	26.17

**Table 3 ijerph-18-04488-t003:** Construct correlations and square roots of AVE.

	BI	IA	IT	IS	MC	PS	PSY
BI	**0.900**						
IA	0.407	**0.883**					
IT	0.092	0.099	**0.852**				
IS	0.516	0.312	0.211	**0.932**			
MC	0.293	0.285	0.157	0429	**0.806**		
PS	0.469	0.347	−0.003	0.440	0.349	**0.747**	
PSY	0.454	0.271	−0.028	0.324	0.166	0.362	**0.772**

Notes: Boldface numbers on the diagonal are the square root of the average variance extracted BI = behavioral intention, IA = information acquisition, IT = information transmission, IS = information selection, MC = message credibility, PS = perceived seriousness, PSY = perceived susceptibility.

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
