# Peer review of "Understanding the Health Behavior Decision-Making Process with Situational Theory of Problem Solving in Online Health Communities: The Effects of Health Beliefs, Message Credibility, and Communication Behaviors on Health Behavioral Intention"

_ijerph, 2021, doi:10.3390/ijerph18094488_

Round 1
Reviewer 1 Report
Review Report for ijerph-1071155
I am glad to have reviewed the manuscript. The research topic is interesting, and the manuscript reads fluently. However, I still have several concerns as discussed below. I hope that the authors can make relevant corrections.
- Use a theoretical framework or theory to support the model. In the current version, the authors provide few literature review or theoretical background information. The authors may try to select a proper theory to support their research model. This helps convince reviewers or readers of the research model and results.
- There are too many hypotheses. First, consider which of them are new and which of them are already tested in prior studies. Then, highlight new hypotheses. Second, use mediation hypothesis to replace some of the current hypothesis. Also, the authors should run some mediation tests to further our understanding of the research model and the relationship among different variables
- I am not sure why the 4.3 is included in the research. Please better connect this section with other parts if the authors hope to keep it. Or the section may be discussed before quantitative results as model free analysis? Or the authors integrate this section with the discussion section.
- There should be some alternative research models. For example, message credibility may serve as moderators. I suggest that the authors can add a post-hoc analysis section to discuss this possible moderation effects. In addition, HPV is mainly for female although male also faces the risk. So if we use gender as moderator, is there any difference in the results?
- Please improve the methodology section. First, add the common method bias test; second, add the non-response bias test, and third, provide the VIF values for independent variables.
Minor:
- It should be better to use Implication for Research as the title of 5.2 since the authors do not use any theory to support their study and it is not easy to convince readers what is the contribution of the current study to theory but not research.
- I suggest that the authors may add a conclusion section. It is not wise to end an article with the limitation section since readers may pay more attention to the opening and ending sections.
Author Response
Thank you very much for your constructive comments for our paper. We have revised our paper item to item based on your comments. Please see the detailed comments in our attached file.

Reviewer 2 Report
The manuscript presents and evaluates research model which uses the opinions and communication habits of participants in relation to forecasting health behaviors in Online health communities.
The problematic covered by the research is very actual, so pertinent and therefore of great interest.
It is well written and structured, however in my opinion the following aspects require to be addressed:
- The abstract needs to be restructured: problem, motivation, aim, methodology followed, main results and their impact is recommended.
- Keywords should be in alphabetical order.
- There is a paragraph missing at the end of introduction about the manuscript organization.
- The authors miss to inform in how the participants were validated, was the survey tool capable of identifying repeated respondents? If so how? The same person can have for example two email addresses just to have a double chance to earn the given incentive.
- Suggestions to overcome the provided limitations require more explanation
Author Response
We appreciate your recognition of your research and acknowledge of our contribution of the research. We also thank you for your constructive comments to improve our paper. We have addressed all comments as in this response letter.
